# Nutrients and the Circadian Clock: A Partnership Controlling Adipose Tissue Function and Health

**DOI:** 10.3390/nu14102084

**Published:** 2022-05-16

**Authors:** Aleix Ribas-Latre, Kristin Eckel-Mahan

**Affiliations:** 1Institute of Molecular Medicine, McGovern Medical School, University of Texas Health Science Center, Houston, TX 77030, USA; aleix.ribas@helmholtz-muenchen.de; 2Helmholtz Institute for Metabolic, Obesity and Vascular Research (HI-MAG) of the Helmholtz Zentrum München at the University of Leipzig, University Hospital Leipzig, D-04103 Leipzig, Germany; 3Department of Integrative Biology and Pharmacology, McGovern Medical School, University of Texas Health Science Center, Houston, TX 77030, USA

**Keywords:** adipose tissue, circadian rhythms, adipose progenitor cells

## Abstract

White adipose tissue (WAT) is a metabolic organ with flexibility to retract and expand based on energy storage and utilization needs, processes that are driven via the coordination of different cells within adipose tissue. WAT is comprised of mature adipocytes (MA) and cells of the stromal vascular cell fraction (SVF), which include adipose progenitor cells (APCs), adipose endothelial cells (AEC) and infiltrating immune cells. APCs have the ability to proliferate and undergo adipogenesis to form MA, the main constituents of WAT being predominantly composed of white, triglyceride-storing adipocytes with unilocular lipid droplets. While adiposity and adipose tissue health are controlled by diet and aging, the endogenous circadian (24-h) biological clock of the body is highly active in adipose tissue, from adipocyte progenitor cells to mature adipocytes, and may play a unique role in adipose tissue health and function. To some extent, 24-h rhythms in adipose tissue rely on rhythmic energy intake, but individual circadian clock proteins are also thought to be important for healthy fat. Here we discuss how and why the clock might be so important in this metabolic depot, and how temporal and qualitative aspects of energy intake play important roles in maintaining healthy fat throughout aging.

## 1. The Adipose Tissue Circadian Clock

The circadian clock is a 24-h time keeping system that exists in almost all cells of the body across terrestrial species and regulates numerous physiological and cellular processes. Disparate biological and physiological processes are under the control of this internal 24-h clock, including the sleep/wake cycle, hormone and neurotransmitter secretion, and glucose and fatty acid metabolism. Single cells ultimately maintain these tissue- and organism-wide rhythms, engaging in strong circadian rhythms in metabolism and gene expression. Like other cell types, the mammalian adipocyte contains a key transcriptional feedback loop driven by the transcriptional activating heterodimer, Circadian Locomotor Output Cycles Protein Kaput (CLOCK) and Aryl Hydrocarbon Receptor Nuclear Translocator Like (ARNTL, also known as BMAL1). Together, these transcriptional activators are necessary for driving cellular circadian rhythms through the partial oscillation of the transcriptome [1,2], but the metabolome [2], proteome [3], and other post-translational events including phosphorylation also maintain robust rhythms in the cell, many of which are indirectly dependent on the transcriptional feedback loop of the core clock [4]. These cellular events translate into tissue-specific rhythms, and ultimately organism-wide rhythms, which are remarkably aligned to maintain energy balance and promote health.

Going into more detail at the molecular level, CLOCK and BMAL1, both bHLH-PAS (basic helix–loop–helix; PER-Arnt-Single) proteins, activate transcription by binding to specific DNA elements, E-boxes, in the promoters of target genes (Figure 1).

These target genes include the regulation of their own negative regulators, the PERiod (*PER*) and CRYptochrome (*CRY*) genes, as well as the nuclear receptor subfamily 1, group D (Nr1d1), also known as reverse erythroblastosis virus alpha (Rev-erbα). On the other hand, CLOCK:BMAL1 also promotes the transcription of the nuclear receptors retinoic acid-related orphan receptor alpha (Rorα), which compete with Rev-erbα for a binding site within the response element (RORE) in the Bmal1 promoter [6], or peroxisome proliferator-activated receptor alpha (Pparα) which in turn can regulate *Bmal1* [7]. Bmal1, Clock, Rorα, Rev-erbα, and the PER and CRY families are known as the core clock genes and are the main regulators of the timekeeping in any given tissue throughout the body. Nonetheless, the CLOCK:BMAL1 heterodimer also promotes the transcription of many other metabolic genes which exert profound tissue-specific regulation, including regulation of different adipose tissue depots across the body [8].

While the lipid droplet size of white adipocytes increases as a result of lipogenesis and fatty acid (FA) uptake (hypertrophy) under conditions of positive energy balance, it decreases in response to lipolysis. Excess hypertrophy can lead to cell lysis, at which time free FAs are released into the bloodstream, promoting ectopic lipid storage in other organs, which is thought to be a primary driver of insulin resistance. Interestingly, the molecular clock core is involved in the regulation of lipogenesis and lipolysis, as many of the key enzymes that are involved in lipolysis and lipogenesis are directly regulated by the CLOCK:BMAL1 transcriptional complex [9,10,11,12,13]. In fact, lipolysis has been reported to be highly rhythmic in WAT, resulting in daily fluctuations of free fatty acids (FA) and glycerol in the bloodstream [9]. Interestingly BMAL1 recruitment to target genes in adipose tissue is remodeled under conditions of obesity, a remodeling which can be reversed by nuclear factor kappa B (NF-κB) inhibition [14]. The clock in adipocytes is highly rhythmic in rodents and humans alike, with similar phase oscillations for the core clock components in brown (BAT), subcutaneous (SAT), and visceral (VAT) adipose tissue [9,10,13,14,15]. Interestingly, these oscillations are autonomous to the fat pad, as adipose tissue explant cultures which are devoid of signals coming from the central clock within the hypothalamic suprachiasmatic nucleus (SCN) can maintain rhythmicity ex vivo for some period of time [12]. Several circadian-disrupted genetic mouse models targeting core clock genes support the importance of the circadian clock in WAT [15]. Generally, the global or tissue-specific absence of one of the core clock genes promotes hypertrophy and ultimately obesity, supporting the notion that an intact clock is necessary to protect against metabolic disorders. Observational studies in humans present a strong association of night shift or rotating shift working (in other words, circadian clock disruption) with increased adiposity, type-2 diabetes, and even cardiovascular disease [16].

Nonetheless, besides hypertrophy, enlarged WAT can also be due to an elevated number of adipocytes comprising the fat pad (hyperplasia), which under prolonged nutrient excess can become hypertrophic due to increased lipogenesis and FA uptake (reviewed in [17]). Though adipogenesis can lead to healthy WAT expansion, there is some evidence that in the context of weight gain, wherein both adipocyte size and number increase, only the former can be reduced by weight loss, and the number of adipocytes remains the same [18]. Prior to differentiation into adipocytes, pre-adipocytes can either proliferate or ultimately enter a senescence phase. Interestingly, we have recently shown that the clock in adipose tissue also plays a role (together with the influence of diet) in the proliferation of pre-adipocytes [13]. However, we will specifically address this finding later when we focus on the clock in adipose progenitor cells.

In addition to clock regulation of APC proliferation, the circadian clock controls rhythmic secretion of some adipokines from fat cells, such as leptin. Leptin is a rhythmic BMAL1-regulated gene [19], the rhythmicity of which can be lost in humans undergoing a constant routine protocol of constant wakefulness [20]. Interestingly, leptin mRNA is elevated in subcutaneous and visceral AT of a mouse with global knockout of the *Clock* gene [13], while adiponectin production is significantly depleted in adipocytes of subcutaneous WAT (unpublished observations). The clock, via PER2, also tightly regulates lipid metabolism in fat, with PER2 inhibiting PPARG transcriptional activity, likely through direct targeting of the S112 site of the PPARG protein [21]. Both lipolysis and free fatty acid release have been shown to be highly rhythmic in WAT [9], the rhythmicity of which is altered in the absence of a functional CLOCK protein. Though these rhythms have been demonstrated in WAT, BAT also maintains strong rhythmic activity of a 24-h nature. Loss of the clock in steroidogenic neurons of the ventromedial hypothalamus increases BAT thermogenesis [22], and prolonged light exposure can attenuate the noradrenergic activation of BAT, causing adiposity due to reduced energy expenditure [23]. In WAT and BAT, oxidation and nutrient utilization are also highly dynamic over 24-h (reviewed in [24]. In summary, WAT and BAT are highly rhythmic in function, which depends on strong 24-h rhythms of individual cell types comprising adipose tissue.

## 2. The Circadian Clock in the Adipose Stromal Cell Vascular Fraction

While a number of studies have addressed aspects of circadian gene regulation or metabolism in whole AT, the number of studies focused on understanding clock function specifically in adipose stromal cell compartments is relatively limited in comparison. However, the current understanding of this AT cellular niche is that either APCs, AEC or infiltrate immune cells express clock genes that are regulated similarly in a feedback loop fashion. Figure 2 summarizes some of the current knowledge on this topic.

### 2.1. The Circadian Clock in Adipose Progenitor Cells (APCs)

In recent years, studies have begun to address the role of the clock specifically in APCs. Interestingly, studies addressing the role of PER family members on APCs regulation reveal that APCs isolated from SAT of m*PER2*^Luc^ reporter mice [25] show robust rhythmicity in *PER2*-driven luciferase expression ex vivo [25] to a higher extent than in fully-differentiated adipocytes [26]. As is the case for mouse ASCs, human adipose-derived stem cells have also been shown to have robust rhythms in circadian gene expression [10]. Similarly, synchronized primary cultures of undifferentiated or adipocyte-differentiated ACSs reveal differences in the period length of the oscillatory clock gene expression determined [27]. Thus, the clock seems to be on a different time signature in APCs when compared to mature adipocytes. In vivo studies where APCs have been isolated at different times of the circadian period have confirmed the robustness of the clock in APCs, revealing rhythms in circadian clock gene expression that are particularly robust for the BMAL1 and PER3 genes [28]. The reason for differences in clock robustness between pre- and fully differentiated adipocytes has not fully been explored experimentally but could reflect changes in cellular demands. Two important processes that APCs engage in include cellular proliferation and differentiation. Interestingly, in the latter, PER3 and CLOCK appear to be have a strong local role attenuating adipogenesis through the transcriptional regulation of the adipogenic gene *Klf15* [28] and the glucocorticoid-induced leucine zipper (GILZ, also known as “delta sleep-inducing peptide immunoreactor”, DSIP) [29] respectively. Deletion of PER3 can promote adipogenesis in vivo by de-repression of the *Klf15* gene. Earlier studies had identified GILZ as a key regulator of adipogenesis in pre-adipocytes cell lines, where it was shown to potently repress transcriptional activation of the PPARγ gene promoter ([30] and reviewed in [31]). In addition, its patterns of expression are food-entrainable and highly dynamic over the course of 24-h in WAT and liver [31]. More studies will shed light on the involvement of other core clock genes in the differentiation of APCs.

Prior to differentiation, APCs proliferate to maintain the necessary pools of APCs within the fat pad [32]. Proliferation itself is to some extent under the direct control of the circadian transcriptional regulators CLOCK and BMAL1 and, consequently, many cell cycle genes are capable of oscillating in a variety of cell types and tissues under physiological conditions [33,34,35,36]. In a recent study addressing rhythmic transcripts in human WAT, an interaction network for cell cycle and centrosome regulation factors were revealed among 727 circadian genes identified [36]. The link between the APC clock and proliferation is not an exception. For example, in a recent study of ours, we revealed that the proliferation of APCs in healthy AT is diurnal in vivo, with peaks in proliferation at the end of the feeding cycle (light onset in nocturnal animals) and a trough at the end of the fasting phase [13]. This diurnal proliferation is dependent on both the cellular clock and rhythmicity in energy intake. Interestingly, both the timing and quality of energy intake can dramatically alter this diurnal proliferation, with fasting and a high fat diet both destroying the rhythmic pattern of proliferation in APCs in both subcutaneous and visceral fat [13]. These separate circadian influences on APC proliferation will be explored in subsequent sections.

While APC proliferation is under control of the circadian clock, the clock is also a critical modulator of adipocyte differentiation though the action of glucocorticoids and other adipogenic hormones. The 24-h oscillation of glucocorticoids is well known, and loss of its circadian regulation is correlated with obesity in humans. One study in particular revealed the critical importance of glucocorticoid rhythms in adipocyte differentiation, revealing that the combination of PPARG accumulation in pre-adipocytes and rhythmic hormonal pulses of glucocorticoid action are required to strictly control the differentiation process [37]. Pulses of glucocorticoid signal that exceeded 24-h resulted in maximal differentiation, suggesting why stress and disorders such as Cushing’s Disease, both of which cause increases and temporally dysregulated glucocorticoid secretion, can contribute to obesity and increases in AT mass.

In summary, APC proliferation and the process by which they differentiate into adipocytes is tightly regulated by the circadian clock. Though the precise direct and indirect mechanisms are not fully understood, rhythmic energy intake appears to be a 24-h process that is tightly linked to these regulatory processes.

### 2.2. Endothelial and Immune Cells of the SVF: An Additional Role for the Clock in Adipose Tissue?

Though the circadian clock appears to contribute to diurnal proliferation in APCs, it may also be an important part of the vascular and immune functions of adipose tissue depots. Like other cell types, endothelial and immune cells show rhythmicity in the body, though less is known from the perspective of their circadian role in the adipose stromal vascular fraction specifically. Nevertheless, there are numerous examples in which the clock in endothelial cells is protective against certain diseases or disorders. For example, vascular PPARG is thought to drive circadian variations in heart rate and blood pressure in a BMAL1-dependent manner [38]. Recently, it was shown that circadian and BMAL1 regulation of the claudin 5 gene (CLDN5), a tight junction protein at the inner blood-retina barrier, protects against retinal pigment epithelium cell atrophy [39]. Interestingly, even the efflux of xenobiotics by the blood brain barrier (BBB) occurs, a process that is largely dependent on rhythmic intracellular magnesium flux in endothelial cells of the BBB. Indeed, this process was shown to be regulated by the BMAL1 target gene, TRPM7, a magnesium transporter. Ablation of *BMAL1* led to a reduction in the number of oscillatory genes in this cell type [40]. Another process where the clock plays a role specifically in endothelial cells includes the blood coagulation process, which is accomplished via the regulation of Thrombomodulin (TM), another oscillatory BMAL1 target gene [41]. Thus, circadian rhythms are present in endothelial cells, and likely play an important role in adipose tissue vascularity as well. Interestingly, in our recent study, which analyzed diurnal proliferation within the SVF as well as in purified APCs, we observed a diurnal pattern of proliferation in isolated CD31-positve cells from the SAT, in contrast to what we observed in VAT [13]. Why this diurnal activity of endothelial cells is depot-specific is not clear, but may reflect the metabolic demands of the vasculature for specific adipose tissue activities.

Immune cells also have been shown to host strong rhythms at both the cellular level, as well as at the organism-wide level. Immune cells have autonomous rhythms, as demonstrated by the ex vivo culture of macrophages from the spleen, which can maintain rhythmic production of TNF-alpha and interleukin 6 (IL-6) [42]. Interestingly, even rhythmic homing to lymph has been observed for specific cell types [43]. Very recently, circadian immune cell function was shown to be important for the clearance of Amyloid-beta 42 (Aβ42). Specifically, using bone marrow-derived macrophages (BMDMs), one study revealed that cell surface proteoglycans (PGs, which were already known to negatively regulate Aβ42 clearance) oscillated over the circadian cycle, with peak levels occurring antiphase to peak levels of Aβ42 phagocytosis [44]. Chemically depleting PG levels blocked the circadian variance in Aβ42 clearance, elevating phagocytosis at the nadir. Thus, the circadian clock has specific cellular mechanisms in immune cells by which it can exert a tissue-specific response.

Not only are there cell autonomous rhythms in endothelial and immune cells independent of their environment, there is also rhythmic communication between endothelial cells and immune cells that can promote the rhythmic behavior of one cell type. For example, adrenergic nerve enervation to target tissues can regulate endothelial cell production of cell adhesion and chemokine factors, which ultimately drives the rhythmic recruitment of leukocytes to target tissues [45]. This time-of-day-dependent migration pattern to specific organs, a property that relies predominantly on the rhythmic expression of pro-migratory factors in endothelial cells, also depends on the autonomous clock residing in leukocytes themselves [46]. Though such mechanisms have not been explicitly demonstrated in the stromal vascular fraction of adipose tissue, it is likely that rhythmic cues from one cell to another promote the rhythmic activity of specific cell types within this important depot. Our data has revealed that under conditions of energy balance, immune (CD45-positive) cells of the visceral fat have a very low level, albeit a diurnal pattern of proliferation [13]. Though immune cell recruitment remains considerably lower in the fat pad of an organism under conditions of energy balance, such activity may be important in response to nutrient challenge, where macrophage recruitment to adipose tissue is elevated.

## 3. Nutrients as Zeitgebers in Adipose Tissue

Since diet itself appears to be an important modulator in adipose progenitor cell proliferation and further differentiation, it is important to consider the extent to which nutrients can alter clock function at the cell- and tissue-specific level. While the number of studies ascertaining the effect of specific food ingredients in the liver clock is extensive, the number of studies focusing in the adipose tissue (AT) clock is limited. Table 1 summarizes reported connections between specific nutrients and the circadian clock in AT.

### 3.1. Effect of High Fat Diet (HFD) in the AT Clock

The high fat diet (HFD), as a mixture of nutrients, has been studied extensively as a modulator of both peripheral and central clocks. However, how specific nutrients or metabolites within HFD or by HFD metabolism affect the clock remains to be fully elucidated. For example, palmitate (converted from palmitic acid) produces phase-shifts in fibroblast Bmal1-driven reporter (Bmal1-dLuc) rhythms in a time-dependent manner, increasing both the period and the amplitude [49]. Interestingly, this effect seems to be cell-specific since the effect is shifted in the corresponding differentiated adipocyte [49]. It is thought that released free fatty acids from mature adipocytes, such as palmitic acid, can also profoundly affect the clock. Examples include docosahexaenoic, eicosapentaenoic and 8-hydroxy-eicosatetraenoic acids, which are natural ligands for PPARα [50]. Monounsaturated fatty acids (MUFAs), docosahexaenoic and eicosapentaenoic acids, as well as arachidonic acid metabolites, are also natural ligands for PPARγ [50]. Likewise, stearic acid is a natural ligand for RORβ [51], and several metabolites derived from cholesterol are natural ligands for RORα and RORγ [51]. Interestingly, while turning on the clock, many of these natural ligands exert a wide range of beneficial metabolic effects stimulating fatty acid oxidation and lowering glucose and insulin plasmatic levels, among others [50,51]. Thus, while HFD has pronounced effects on the circadian clock in specific tissues [71], specific metabolites within the diet and as a result of HFD metabolism in vivo can also modulate clock function in a cell- and tissue-specific way. One effect of this may be the circadian ‘misalignment’ observed across tissues of the body under prolonged HFD feeding, where metabolite rhythms are almost entirely absent in the circulation [47].

Even circadian behavior is greatly altered by HFD consumption. For example, male mice fed a HFD ad libitum increase their caloric consumption during the 12-h light phase, concomitant with a circadian period lengthening compared to control chow-fed animals [48]. This increase in diurnal food intake seems to be gender-dependent since Yanagihara et al. reported that C57BL/6J female mice fed a HFD (40% Kcal from fat) for six weeks did not show increased diurnal food intake, but developed obesity, hyperlipidemia, and hyperglycemia [72]. While inducing obesity in both males and females after six weeks on HF [48,72], the increased caloric intake in males during the rest phase results in alterations in the diurnal patterns of circulating metabolic markers, together with a reduced amplitude in the expression of *Clock*, *Bmal1*, and *Per2* in the AT and the liver [48]. Females not displaying a change in the temporal aspects of food intake presented minimal changes in rhythmic mRNA expression of clock genes in peripheral tissues [72]. Interestingly, mice fed a HFD during the light period gain significantly more body weight than mice fed the same diet with the same amount of calories during the dark period [54]. Thus, the timing of food intake, particularly in the case of a high caloric diet enriched in fat, is important for energy balance and circadian behavior. Restricting HFD to the dark period as opposed to ad libitum access results in lower body weight and restored glucose tolerance and diurnal rhythms in metabolic regulators [55], while restricted HFD toward the end of the active period results in increased adiposity when compared with mice fed a similar diet in the beginning of the active period [73]. The central clock in the SCN ultimately dictates the food intake pattern and clearly responds to a HFD accordingly [48], nonetheless, a defect in the molecular clock specifically in the AT also affects the food intake pattern and consequently promotes fat expansion even on a regular diet [74]. Thus, although the disruption of the clock itself is sufficient to promote an increase in body weight even in the absence of a HFD [75], HFD augments the phenotype, promoting AT enlargement either through adipocyte hypertrophy or hyperplasia [32,76]. Though the creation of new adipocytes by hyperplasia leads to healthier fat in the short term, prolonged nutrient excess leads to not only the formation of new adipocytes, but their ultimate enlargement via hypertrophy. Lineage tracing studies have confirmed that HFD contributes to WAT hyperplasia by the induction of PDGFRα+ cells [77], often considered a marker of APCs. Yet, the diurnal control of APC proliferation seen in lean mice under physiological conditions is completely lost under HFD [13], highlighting once more the importance of the AT clock in the etiology of obesity and its response to nutrient challenge. More studies using specific micronutrients within a HFD will be necessary to understand what other components function as key zeitgebers for the AT clock.

### 3.2. Effect of Sugar on the AT Clock

Even in the absence of excess fat in the diet, sugar has a clear effect on the AT clock. Sugar as a whole and more specifically, fructose, has similar effects in terms of adiposity [52,53]. Although how sugar and derivates specifically affect the AT clock is not fully understood, the timing of consumption brings phenotypical changes in rodent models [52,53]. As is the case for a HFD, restricting the access of sugar [52] and fructose [53] to the active/dark phase in rodents improves metabolism concomitant with a reduction in adiposity. In fact, it is known that our predisposition to metabolize glucose is time-dependent, as the pancreatic function responding to systemic glucose levels releasing insulin tends to be more efficient during our active phase (light period for humans and dark period for rodent models) [78]. Molecular mechanisms behind this include transactivation by the CLOCK:BMAL1 heterodimer at specific target genes involved in glucose uptake and metabolism. This transactivating capacity can be compromised by an increase in NAD(P)H levels as a result of an increase of fatty acids and glucose in blood after ingesting simple sugars [79]. SIRT1, a nicotinamide adenine dinucleotide (NAD^+^)-activated deacetylase for BMAL1 among other targets, can inhibit CLOCK:BMAL1 activity [80]. Furthermore, other proteins involved in the regulation of the glucose homeostasis, including PEPCK and PDK4, are regulated by the clock genes CRY1 and CRY2 as well as by the rhythmic metabolic genes PPARα and PPARγ, providing additional connections between the molecular clock and regulation of glucose metabolism [81,82].

### 3.3. Effect of Polyphenols in the AT Clock

Despite not providing calories as is the case for fat and sugar, polyphenols, naturally found in plants, possess not only a wide range of beneficial effects for human health, but function as circadian clock modulators in vivo [83]. Thousands of polyphenols exist, and include compounds such as flavonoids, polyphenolic amides, and phenolic acids, which collectively can be found in food products such as spices, dried herbs, berries, nuts and seeds. Although their effects on the clock are not limited to the adipose tissue, resveratrol, proanthocyanidins (PAs), and epigallocatechin gallate (EGCG) have also been shown to modulate the AT clock. Resveratrol is thought to increase SIRT1 activity through a direct allosteric activation of its N-terminal domain [59], and can reverse AT circadian gene expression changes induced by a HFD. This is accompanied by a reduction in body weight and fat mass [57]. Interestingly, resveratrol administration to gray mouse lemurs can adjust circadian rhythms of both locomotor activity and body temperature [58]. Similarly, PAs also modulate the expression level of clock-core and clock-controlled genes in the mesenteric AT of healthy and obese rats, while affecting to some extent the acetylated protein ratio of BMAL1 [60]. Although not directly studied in AT, it has been shown that PAs activates SIRT1 in the liver, reducing cholesterol and triglycerides hepatic content [84] in turn. Using metabolomics, Aragonès et al. found that PAs increase the hepatic NAD^+^ content in a dose-dependent manner by specifically modulating the hepatic concentrations of the major NAD^+^ precursors and the mRNA levels of genes encoding the enzymes involved in the cellular metabolism of NAD^+^ [84]. More importantly, when PAs are administered in rats acutely, it modulates the hepatic levels of NAD^+^, the acetylated protein levels of BMAL1, and the gene expression levels of nicotinamide phosphoribotransferase (NAMPT) (a BMAL1 target gene, which is the rate-limiting enzyme in the NAD^+^ salvage pathway) in a time-dependent manner [85]. Thus, as the case for resveratrol, it cannot be ruled out that SIRT1 is playing a role in the functionality of the BMAL1:CLOCK core clock in AT as a result of PAs consumption. The protein levels of SIRT1 in AT (either WAT or BAT) are also modulated by another type of polyphenol. Chronic treatment of EGCG for 16 weeks increases SIRT1 levels in AT of mice fed a high fat and high fructose diet (HFFD) to comparable levels as lean mice [61]. Importantly, the restored levels of SIRT1 in AT are accompanied by a general metabolic improvement (e.g., loss in body weight, improvement in glucose and insulin sensitivity, reduction in cholesterol and triglycerides plasmatic levels) and a gain of BMAL1 and SIRT1 gene expression oscillation in BAT [61]. More studies will be necessary to assess whether acute treatment of EGCG also modulates the clock in WAT and if time of administration matters. Studies determining the effect of PAs in the liver [85] and the hypothalamic [86] clock concluded that the precise time of administration controls the effect of this polyphenol on the molecular clock in vivo, suggesting that PAs (and potentially other polyphenols including resveratrol and EGCG), could be non-photic signals to entrain the clock in WAT.

### 3.4. Effect of Natural Alkaloids and Passionflower Extract on the AT Clock

Similar to polyphenols, natural alkaloids including caffeine, harmine and lycorine are also present in high abundance in plants. While the effect of polyphenols on the AT clock has been shown through numerous in vivo studies, the effects of these specific natural alkaloids and the passionflower extract on the adipocyte clock have been investigated both in vivo and also using an in vitro model, the NIH3T3 cell line. Caffeine significantly lengthens the period of mice based on actigraphy, suggesting a direct or indirect effect on the central clock [63]. In the same study, a similar effect was observed on the AT peripheral clock using NIH3T3-mBmal1-Luc cells as a model mouse, which are fibroblasts stably expressing a vector harboring the murine Bmal1promotor (−611 to +154) [63]. Using a similar model (NIH3T3 cells containing the luciferase reporter gene driven by the BMAL1 promoter region (−197 to +27)), another study showed how harmine [64] and lycorine [65] lengthened the circadian period of BMAL1 transcription. While the effect of harmine is mediated by an enhancement of the nuclear translocation of RORα [64,66], the mechanism in the case of lycorine is unclear, but the authors propose that it could be distinct from that of harmine [65]. For the case of the passionflower extract treatment, the molecular clock in NIH3T3 is modulated via an induction of high-amplitude rhythms in the expression of the negative clock regulators PER2 and CRY1 [66].

With the exception of lycorine, the toxicity of which prevents its use as a medicinal, it is important to note that studies addressing the effect of natural alkaloids and plant extracts in the molecular clock may reveal tractable therapeutic options for treating unhealthy AT and improving whole body metabolism. Interestingly, 56 of the top 100 best-selling drugs in the United States, including the top seven, target the product of a clock gene [11]. Though chronotherapy using approved drugs is a rapidly growing area of needed study, understanding how nutrients function as zeitgebers to improve health or prevent disease will be important as more natural compounds are found to have zeitgeber properties on the clock in AT.

### 3.5. Effect of Endogenous Biomolecules in the AT Clock

From a molecular point of view, retinoic acid and heme are known clock modulators due to their ability to bind to RORα [70] and REV-ERBα/β [69] receptors, respectively, which have opposite transcriptional effects at the BMAL1 promoter, ultimately affecting the circadian clock in tissue-specific ways [11]. Interestingly, REV-ERBα regulates fibroblast growth factor 21 (FGF21) signaling uniquely in adipose tissue compared to liver, via pronounced repression of its receptor, the βKlotho gene [87]. In humans, WAT heme levels are positively correlated with the expression of heme biosynthesis-related genes, some of which increase in expression parallel to adipogenic genes during adipogenesis [88]. The blockade of heme synthesis blocks adipogenesis, but less understood is whether heme may also alter adipogenesis and adipose function through the clock. In addition to the REV-ERBα/β nuclear receptors, heme serves as a ligand for the neuronal PAS domain protein 2 (NPAS2), a CLOCK homolog highly expressed in the forebrain that can interact with BMAL1 [89]. There are several points of potential interaction between dietary iron (of which the heme form is most easily absorbed) and the clock. Serum ferritin in humans is negatively associated with serum leptin in patients with metabolic syndrome, an association which is supported by mice fed a high-iron diet [90]. This relationship was found to be caused by iron-induced suppression of cAMP-responsive element binding protein (CREB), a transcription factor which, like BMAL1, regulates leptin gene expression. CREB is also a primary driver of PER1 expression, its phosphorylation being involved in light-induced phase shifts in clock function [91]. Additional studies will be needed to further understand the extent to which AT function might be altered in vivo by dietary iron in a clock-dependent manner.

Interestingly, in mouse fibroblast NIH3T3 cells, retinoic acid has been shown to upregulate *PER1* and *PER2* expression in an E-box-dependent manner [69]. On the other hand, rather than directly modifying the CLOCK:BMAL1 heterodimer, endogenous polyamines positively affect the efficiency of the PER2:CRY1 interaction both in the liver and in NIH3T3 cells [68]. Furthermore, a more recent study has shown that polyamines also induce the protein synthesis of BMAL1 and REV-ERBα, while shortening the period of the clock in NIH3T3 cells [67]. Both studies are consistent with the shortening effect of polyamines on the clock of NIH3T3 cells [67,68]. A similar effect on period length has been observed in vivo in a mouse model, where supplementation of polyamines in the drinking water of aged mice can restore normal periodicity [68].

### 3.6. Effect of Xenobiotic Metabolism and Other Endogenous Compounds on the AT Clock

Though the described molecules and nutrients provide zeitgeber properties for numerous peripheral tissues, xenobiotic metabolism, and other endogenous molecules such as bile acids, they also have circadian clock-altering effects. In adipose tissue specifically, our studies have revealed that fasting or isocaloric feeding that extends through the 24-h cycle as opposed to being restricted to the active phase or ad libitum abolishes the diurnal variance in APC proliferation [13]. Time-of-day nutrient digestion has also proved to be important for rhythmicity in metabolic gene expression in other tissues. Time restricted feeding (TRF) under both chow and high fat diet conditions can boost the amplitude of rhythms in hepatic gene expression and in serum metabolite abundance [55,92,93,94,95,96], while restricting feeding to the rest phase instead of the active phase puts time-keeping between peripheral clocks and the central pacemaker out of sync [97]. Restricted feeding to the rest phase also destroys the temporal pattern of circulating bile acids in the serum. Serum bile acids come from the conversion of cholesterol in the liver and ultimately control body-wide triglyceride levels in a circadian way. This depends on the rhythmic activation of several nuclear factors, including the farnesoid X receptor (FXR), small heterodimer partner (SHP), and sterol regulatory element binding protein 1 (SREBP). Though the effects of TRF have been more heavily studied from the perspective of liver gene expression and function, delayed breakfast accumulation has been shown to delay PER-responsive gene transcription in human adipose tissue [98], and TRF can also partially restore gene expression rhythms in WAT under constant light conditions [99]. Even under high fat diet conditions, TRF can reduce fat accumulation and crown-like structures in epididymal WAT, suggestive of reduced inflammation and immune cell infiltration into WAT [93]. In addition, TRF under the conditions under high fat diet feeding can reduce the whitening of brown fat [92]. As feeding time and quality can dramatically alter serum bile acid patterns, resulting changes in circulating bile acids and gut microbiota may have a pronounced effect on AT as a result. Interestingly, one recent study using a mouse model of type II diabetes revealed that feeding mice a resveratrol-rich diet resulted in enhanced BAT activity and WAT browning, predominantly due to the remodeling of the gut microbiota, as antibiotics partially reduced the beneficial effects of the resveratrol-rich diet [100]. Thus, the xenobiotic metabolism likely works through multiple circadian-dependent mechanisms to alter AT function throughout the body.

Though this review centers predominantly on nutritional and circadian regulation on AT properties, not to be dismissed is the role that time plays on AT biology through the process of aging. Aging of AT involves not only a pro-inflammatory profile, but also cellular senescence in adipose progenitors (reviewed in [15,101]. As circadian rhythms are known to decline in robustness throughout aging, studies have sought to address how amplifying circadian rhythms might delay metabolic disease and preserve tissue-specific function. One mechanism of circadian potentiation involves the small molecule Nobiletin (NOB), which activates ROR nuclear receptors. Recent studies have shown that NOB promotes healthy aging in part by augmenting mitochondrial activity in skeletal muscle [102]. As part of this study, NOB was shown to reduce the high fat diet-induced enlargement of BAT. Similarly, NOB was previously shown to benefit metabolic activity in young mice under a high fat diet, where NOB reduced fat mass and white adipocyte cell size in vivo [62]. Molecules such as resveratrol are thought to prevent aging in part through their activation of the sirtuin protein SIRT1 [103]. Circadian activity declines in the SCN of aged mice, but activation of SIRT1 has been shown to prevent this decline in circadian function, in part by a previously identified circadian clock transcriptional mechanism that controls the expression of a rate-limiting enzyme involved in NAD^+^ synthesis [80,104,105]. Resveratrol is known to increase BAT thermogenesis, and substantially decrease WAT mass in mice even when fed vivarium chow [106]. Interestingly, under paradigms of sleep fragmentation in mice, resveratrol treatment can attenuate inflammation within VAT as well as insulin resistance, reinforcing the idea that mechanisms which are known to prevent aging and metabolic disease involve pronounced changes in AT function. Further studies using AT-specific clock deletion or, alternatively, AT-specific clock reconstitution in circadian mutant mouse models will aid in our understanding of how much of age- and circadian-based metabolic decline involves rhythms in WAT specifically.

## 4. Conclusions

In summary, the circadian clock in adipose tissue is a robust, highly complex system that can respond to nutrient cues (Figure 3) in specific ways. The clock genes in particular appear to have complex roles in adipose tissue function, though at the level of CLOCK and PER protein activity, the circadian clock is capable of functioning as a break for adipogenesis, which is likely critical for controlling AT expansion throughout the lifespan, but could be harmful in the context of nutrient excess. Though clock genes appear to be important regulators of adipose tissue function and expansion, so too is rhythmic feeding and nutrient quality. As more nutrients and metabolites are identified to harbor *zeitgeber* properties for the AT clock, a better understanding of how diet and circadian lifestyle can work together to promote AT health throughout aging will likely emerge.

## Figures and Tables

**Figure 1 nutrients-14-02084-f001:**
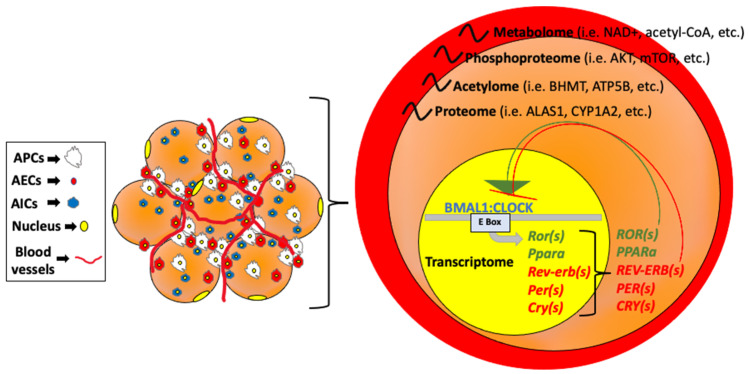
Circadian regulatory factors in cells of the body. Left: Adipose tissue, which is comprised of adipocyte progenitor cells (APCs), endothelial cells (AECs), and immune cells (AICs) contains the same clock regulatory factors important for rhythmicity in other cell types. Right: The PAS-domain containing transcription factors CLOCK and BMAL1 heterodimerize to form a complex that binds to and generally activates target genes harboring E box consensus sites in the promoter. Target genes include the negative regulators of the complex, including the *PER* and *Cry* genes, which when rhythmically expressed, directly interact with the CLOCK:BMAL1 heterodimer and block subsequent transactivation of target genes. Rev-erbα and Rev-erbβ can transcriptionally repress BMAL1 expression, while Rorα and Pparα can transcriptionally activate BMAL1. This transcriptional feedback loop regulates the rhythmicity of hundreds of genes involved in metabolism (resulting in rhythmic metabolite production [2] among other rhythmic cellular processes). However, post-transcriptional and post-translational modifications are also highly dynamic throughout the 24-h cycle. These modifications include acetylation and phosphorylation of proteins within the cell [4,5].

**Figure 2 nutrients-14-02084-f002:**
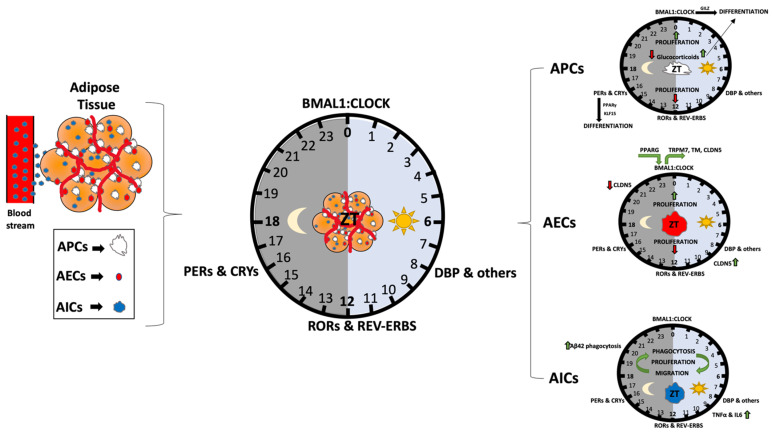
Circadian regulation of adipose tissue and its resident cells. Adipose tissue composition includes mature adipocytes (yellow), and cells of the stromal vascular fraction (SVF), which include immune cells (AIC), endothelial cells (AEC) as well as adipocyte progenitor cells (APCs), which can undergo adipogenesis to become mature adipocytes. Collectively, these cells are controlled in part by circulating factors that access adipose tissue via the vasculature (**left**). Specific circadian cues known to be important for adipose tissue biology include glucocorticoids and PPARγ (which regulate pre-adipocyte differentiation), as well as clock genes themselves, which can control proliferation and function of specific SVF cells through regulation of CLOCK:BMAL1-target genes (**right**). The BMAL1-target gene CLDN5, a tight function protein expressed in endothelial and epithelial cells, is known to be rhythmic and important for some microvascular function. In addition to immune cell regulation by clock genes, the proinflammatory cytokines, TNFα and IL6, are also clock-controlled, and can modulate the activity of immune cells such as macrophages.

**Figure 3 nutrients-14-02084-f003:**
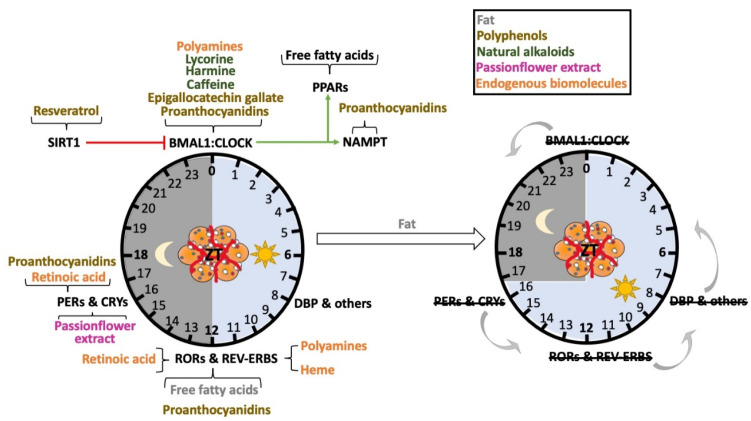
Numerous nutrients function as zeitgebers for the adipose tissue clock. Many xenobiotics and endogenous nutrients can directly or indirectly alter clock function via regulation of circadian clock genes (**left**). Specific molecules reported to affect unique clock genes are shown on the left. Excess fat in the diet appears to not only cause a phase advance in the core clock genes (**right**, gray arrows to the left), but also causes changes in period length. These observations come in part from the use of circadian clock gene reporter mice, which can be analyzed for the circadian phase using bioluminescence.

**Table 1 nutrients-14-02084-t001:** Examples of nutrient components on the circadian clock across tissues.

Nutrient/Dietary Components	Examples	Effect on the AT Clock	References
Fat	Whole diet	Circadian period lengthening, accompanied by an increase in food intake during the light phase and a change in AT clock gene expression in male mice, resulting in a misalignment of the circadian clock across tissues of the body.	[2,47,48]
	Ingredients: Palmitate	Clock modulation in undifferentiated and differentiated NIH3T3 cells.	[49]
	Fat derivates: Free fatty acids	Natural ligands for PPARα, PPARγ and RORs, directly affecting clock gene expression	[50,51]
Sugar	30% sugar dissolved in water	Restricted access during the dark phase limits the body weight gain in rats	[52]
	Fructose	Restricted access during the dark phase limits the AT expansion in mice	[53]
Restricted access	Western diet	Restricted access during the dark phase limits the body weight gain and AT expansion in rodents, while restores the glucose tolerance and diurnal rhythms of metabolic regulators	[54,55,56]
	Chow diet	Restricted access during the dark phase limits the body weight gain and AT expansion in rodents	[52,53]
Polyphenols	Resveratrol	Increases SIRT1activity, modulates circadian rhythms of locomotor activity and body temperature, and reverse adipose tissue-specific circadian gene expression changes induced by a high fat diet	[57,58,59]
	Procanthocyanidins (PAs)	Modulate the expression of clock-core and clock-controlled genes in mesenteric AT of healthy and obese rats, including BMAL1, Clock, Rorα, Rev-erbα, PER2 and Nampt.	[60]
	Epigallocatechin gallate (EGCG)	Restores BMAL1 rhythmic expression in BAT concomitant with a metabolic improvement in DIO mice	[61]
	Nobiletin (NOB)	Reduces fat mass and white adipocyte cell size in vivo	[62]
Natural alkaloids	Caffeine	Lengthens the period of BMAL1 expression in NIH3T3 cells	[63]
	Harmine		[64]
	Lycorine		[65]
Passionflower extract		Induces high-amplitude rhythms in the expression of PER2 and CRY1 in NIH3T3 cells	[66]
Endogenous biomolecules (metabolites, precursors)	Polyamines	PERiod shortening and stimulation of BMAL1 and Rev-erbα synthesis in NIH3T3 cells	[67,68]
	Retinoic acid	Ligand for RORα. Induction of PER1 and PER2 expression in NIH3T3 cells	[69,70]
	Heme	Ligand for REV-ERBα	[69]

## Data Availability

Not applicable.

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
