# Peer review of "Nutrients and the Circadian Clock: A Partnership Controlling Adipose Tissue Function and Health"

_nutrients, 2022, doi:10.3390/nu14102084_

Round 1
Reviewer 1 Report
Aleix Ribas-Latre et al review addresses an original topic such as the participation of the circadian clock on adipose tissue function, so it is very interesting. However, some points should be improved before acceptance for publication.
- It would be important to add/extend a general description of the circadian clock function, which would facilitate reading for people who are not specialists in the subject. Schema could even be added.
- In the same sense, the legends of the diagrams should be more detailed to facilitate their understanding.
- In general, in this review it is understood that adipogenesis/lipogenesis as causes of hypertrophy. In fact, adipogenesis by itself does not generate adipocyte hypertrophy, it is even considered a protective mechanism against adipocyte hypertrophy, contributing to what is called healthy expansion of adipose tissue. hypertrophy is a consequence of positive energy balances and increased lipogenesis. These concepts should be reviewed so as not to lead the reader to wrong conclusions.
- The relationship between the circadian clock and the pattern of adipokine secretion should be discussed since it is a characteristic of adipocyte dysfunction.
- Finally, it is not clear from all partial conclusions whether the effects of circadian rhythm components produce beneficial or detrimental effects on adipose tissue function.
Author Response
Reviewer #1: Aleix Ribas-Latre et al review addresses an original topic such as the participation of the circadian clock on adipose tissue function, so it is very interesting. However, some points should be improved before acceptance for publication.
We thank the reviewer for reviewing our manuscript carefully and providing their constructive comments. We have significantly revised our manuscript as suggested. All revisions are in alternate font colors in the main text. We have addressed each comment as follows:
- It would be important to add/extend a general description of the circadian clock function, which would facilitate reading for people who are not specialists in the subject. Schema could even be added.
Thank you for this suggestion. We have added a figure (now figure 1) listing some of the key clock components- specifically the core transcriptional feedback loop. We have also listed a few of the post-transcriptional and metabolic oscillations important for cell- and ultimately tissue-specific rhythms to provide readership with less familiarity with the clock system additional information about the cellular 24-hr. clock.
- In the same sense, the legends of the diagrams should be more detailed to facilitate their understanding.
Thank you for this good suggestion. We have expanded each of the figure legends to provide more detail pertaining to specific parts of the figures. The new text is highlighted in red.
- In general, in this review it is understood that adipogenesis/lipogenesis as causes of hypertrophy. In fact, adipogenesis by itself does not generate adipocyte hypertrophy, it is even considered a protective mechanism against adipocyte hypertrophy, contributing to what is called healthy expansion of adipose tissue. hypertrophy is a consequence of positive energy balances and increased lipogenesis. These concepts should be reviewed so as not to lead the reader to wrong conclusions.
We thank the reviewer for highlighting our lack of clarity in this area, and completely agree that as written, the article suggested that adipogenesis was also a primary contributor to unhealthy fat. We have clarified that lipogenesis and hypertrophy are the primary attributes of unhealthy fat. However, we do mention studies suggesting that in the context of weight gain, the increase in adipocyte number cannot be reduced even after weight loss, though hypertrophy can be reversed. The new text summarizing these points can be found in lines 76-80 and also in lines 100-104.
- The relationship between the circadian clock and the pattern of adipokine secretion should be discussed since it is a characteristic of adipocyte dysfunction.
We appreciate the reviewer bringing up this omission in the text. We have now discussed two different adipokines as well as their proposed relationship to the circadian clock. We discuss BMAL1 regulation of leptin, but also data showing altered adiponectin in the context of circadian clock mutant mice. The new text related to adipokine release can be found in 110-116.
- Finally, it is not clear from all partial conclusions whether the effects of circadian rhythm components produce beneficial or detrimental effects on adipose tissue function.
We appreciate the reviewer bringing up this point. We have added some text in the conclusion that pertains to this point. Specifically, though the clock appears to have complex roles in adipose tissue, at least CLOCK and PER proteins appear to have an inhibitory effect on adipogenesis, which is likely important for maintaining adipose tissue mass throughout aging. However, we hypothesize that this restraint could be harmful in the context of nutrient excess. This discussion can be found in lines 504-508.
Reviewer 2 Report
General Comments
This is a well written review that highlights the importance of nutrition/nutrients in adipose tissue circadian biology. The authors cite their own work as well as others referring to both in vitro and in vivo based studies using primarily murine and human systems. Several general elements merit additional attention in addition to the specific comments cited below.
- There is considerable evidence that metabolic pathways, particularly involving oxidative phosphorylation and mitochondrial/TCA cycles are prominently displaying a circadian rhythmicity in multiple adipose depots and liver. While the authors particularly highlight the impact of circadian mechanisms on proliferation as a pathway, devoting some attention to metabolism “writ large” might be helpful for those readers who are just becoming introduced to this topic.
- Time impacts adipose circadian biology through an avenue other than the 24 hour clock cycle, namely, through the process of biological aging. There are publications on this relationship in the literature and the authors should incorporate the impact of aging on adipose circadian mechanisms into their review for the sake of completeness.
- There is a body of literature linking circadian mechanisms to xenotoxicity of nutrients ingested from the environment as well as endogenous compounds (bile salts for example) generated through metabolism. While much of the work has appeared in the liver rather than adipose tissue, it merits inclusion to a review regarding nutrients since time of day of ingestion can impact the metabolic modification of a chemical, both good and bad, for the individual.
Specific Comments
Ln 111. Change “hasn’t” to “has not”.
Ln115. The authors cite reference 21 in the context of GILZ with respect to circadian biology. Studies referring to GILZ by its alternative nomenclature, delta sleeping inducing peptide, had linked circadian mechanisms to its adipogenic expression profile nearly a decade before the publication of ref 21 in 2018. The authors are encouraged to include additional references to provide the appropriate context for the relationship between less well characterized circadian regulatory proteins and adipocyte differentiation.
Ln 163. The sentence beginning “Other process” should be re-phrased possible as “Another process”?
Ln 167. The phrase “adipose vascular” possibly should be rephrased as “adipose vascularity”?
Ln 193. Change “haven’t” to “have not”.
Ln 205. The phrase “significantly extensive” is unclear and should be re-phrased as it implies a statistical relationship.
Table 1. There is considerable literature evaluating the effect of temporally restricted dietary access in rodents on circadian gene expression. It is recommended that the Table reflect this category of studies in addition to those evaluating specific nutrients in the context of the review.
Ln 245, Ref 58. In the context of this murine study, it would be helpful to also cite the work relating to night eating behaviors original reported by Stunkard AJ and colleagues at University of Pennsylvania to provide a human disease equivalent. This would be valuable to give readers an appreciation of the medical consequences. Likewise, this relevant to statements made regarding sugar as a nutrient in Ln 269.
Ln 304. Change “PAs is” to “Pas are”.
Section 3.5, Ln 350. The authors are to be congratulated on their comprehensive review and thorough evaluation of the scope of the field and their attention to Rev-erb and ROR agonists. Related to this, is there any literature specifically focusing on heme or heme containing foods in the context of circadian mechanisms in adipose tissue? Are there links between Rev-erb ligand activation and metabolic consequences on obesity? Even if such literature does not exist, might such a mechanism have a relationship to the ingestion of red meat in the diet and its consequences on adiposity? Would it be worthwhile to speculate about such a relationship to encourage readers to consider such a mechanism for future investigation?
Author Response
This is a well written review that highlights the importance of nutrition/nutrients in adipose tissue circadian biology. The authors cite their own work as well as others referring to both in vitro and in vivo based studies using primarily murine and human systems. Several general elements merit additional attention in addition to the specific comments cited below.
- There is considerable evidence that metabolic pathways, particularly involving oxidative phosphorylation and mitochondrial/TCA cycles are prominently displaying a circadian rhythmicity in multiple adipose depots and liver. While the authors particularly highlight the impact of circadian mechanisms on proliferation as a pathway, devoting some attention to metabolism “writ large” might be helpful for those readers who are just becoming introduced to this topic.
We appreciate the reviewer highlighting the extent to which mitochondrial function also plays a role in circadian-regulated adipose tissue metabolism. We agree that this is needed for the review. We have added additional text related to this topic in lines 117-127. In addition, we have commented on the rhythmic pattern of metabolite oscillations in figure 1 (in response to both reviewers).
2. Time impacts adipose circadian biology through an avenue other than the 24 hour clock cycle, namely, through the process of biological aging. There are publications on this relationship in the literature and the authors should incorporate the impact of aging on adipose circadian mechanisms into their review for the sake of completeness.
We appreciate the reviewer mentioning the concept of age-associated decline with adipose function, particularly because the circadian clock is known to decline throughout aging. We have added a paragraph addressing age- and circadian-associated alterations in metabolic health, with an emphasis on adipose tissue biology. These changes can be observed in lines 476-501.
3. There is a body of literature linking circadian mechanisms to xenotoxicity of nutrients ingested from the environment as well as endogenous compounds (bile salts for example) generated through metabolism. While much of the work has appeared in the liver rather than adipose tissue, it merits inclusion to a review regarding nutrients since time of day of ingestion can impact the metabolic modification of a chemical, both good and bad, for the individual.
We appreciate this comment by the reviewer. We have added information related to time-restricted feeding (TRF) on circadian rhythms in metabolic tissues. We also discuss other endogenous molecules, including microbiota-related metabolites and bile acids in regulation of clock function in peripheral tissues, highlighting their potential role in adipose tissue. This new information can be found in 444-475.
Specific Comments
Ln 111. Change “hasn’t” to “has not”.
We have fixed this typo. Thank you.
Ln115. The authors cite reference 21 in the context of GILZ with respect to circadian biology. Studies referring to GILZ by its alternative nomenclature, delta sleeping inducing peptide, had linked circadian mechanisms to its adipogenic expression profile nearly a decade before the publication of ref 21 in 2018. The authors are encouraged to include additional references to provide the appropriate context for the relationship between less well characterized circadian regulatory proteins and adipocyte differentiation.
We appreciate the reviewer highlighting these early studies. We now cite a much earlier study showing that DISP/GILZ inhibits adipogenesis, as well as another article which demonstrates its food-entrainable oscillations of GILZ in WAT and liver. We thank you reviewer for commenting on these valuable studies. This added text can be found within lines 166-173.
Ln 163. The sentence beginning “Other process” should be re-phrased possible as “Another process”?
We have fixed this typo. Thank you.
Ln 167. The phrase “adipose vascular” possibly should be rephrased as “adipose vascularity”?
We have fixed this typo. Thank you.
Ln 193. Change “haven’t” to “have not”.
We have changed this, as suggested. Thank you.
Ln 205. The phrase “significantly extensive” is unclear and should be re-phrased as it implies a statistical relationship.
We have changed this sentence. Thank you for pointing out the confusing nature of this.
Table 1. There is considerable literature evaluating the effect of temporally restricted dietary access in rodents on circadian gene expression. It is recommended that the Table reflect this category of studies in addition to those evaluating specific nutrients in the context of the review.
We thank the reviewer for pointing out this omission. We have added time restricted feeding (TRF) to the table, including references for both TRF of chow and TRF of high fat, so-called “western diet”. We have added several references pertaining to the effects of TRF on adipose tissue specifically. Thank you for this recommendation.
Ln 245, Ref 58. In the context of this murine study, it would be helpful to also cite the work relating to night eating behaviors original reported by Stunkard AJ and colleagues at University of Pennsylvania to provide a human disease equivalent. This would be valuable to give readers an appreciation of the medical consequences. Likewise, this relevant to statements made regarding sugar as a nutrient in Ln 269.
Ln 304. Change “PAs is” to “Pas are”.
We have fixed this subject/verb agreement. Thank you.
Section 3.5, Ln 350. The authors are to be congratulated on their comprehensive review and thorough evaluation of the scope of the field and their attention to Rev-erb and ROR agonists. Related to this, is there any literature specifically focusing on heme or heme containing foods in the context of circadian mechanisms in adipose tissue? Are there links between Rev-erb ligand activation and metabolic consequences on obesity? Even if such literature does not exist, might such a mechanism have a relationship to the ingestion of red meat in the diet and its consequences on adiposity? Would it be worthwhile to speculate about such a relationship to encourage readers to consider such a mechanism for future investigation?
We thank the reviewer for suggesting this interesting topic. We have described several links between dietary iron, heme, and circadian transcription factor function (CREB, NPAS2, and REV-ERBs) in leptin expression but also in adipogenesis. This additional text can be found in lines 413-431.